# Cryo-EM analysis of the HCoV-229E spike glycoprotein reveals dynamic prefusion conformational changes

Xiyong Song [1,2,3,10], Yuejun Shi[1,4,10], Wei Ding[5,10], Tongxin Niu[6], Limeng Sun[1,4], Yubei Tan[1,4], Yong Chen[2,3], Jiale Shi[1,4], Qiqi Xiong[1,4], Xiaojun Huang [6], Shaobo Xiao [1,4], Yanping Zhu[2], Chongyun Cheng[2], Zhen F. Fu [1,4,7], Zhi-Jie Liu [8,9 ✉] & Guiqing Peng [1,4 ✉]

Coronaviruses spike (S) glycoproteins mediate viral entry into host cells by binding to host receptors. However, how the S1 subunit undergoes conformational changes for receptor recognition has not been elucidated in *Alphacoronavirus*. Here, we report the cryo-EM structures of the HCoV-229E S trimer in prefusion state with two conformations. The activated conformation may pose the potential exposure of the S1-RBDs by decreasing of the interaction area between the S1-RBDs and the surrounding S1-NTDs and S1-RBDs compared to the closed conformation. Furthermore, structural comparison of our structures with the previously reported HCoV-229E S structure showed that the S trimers trended to open the S2 subunit from the closed conformation to open conformation, which could promote the transition from pre- to postfusion. Our results provide insights into the mechanisms involved in S glycoprotein-mediated *Alphacoronavirus* entry and have implications for vaccine and therapeutic antibody design.

[1] State Key Laboratory of Agricultural Microbiology, College of Veterinary Medicine, Huazhong Agricultural University, Wuhan, China. [2] National Laboratory of Biomacromolecules, Institute of Biophysics, Chinese Academy of Sciences, Beijing, China. [3] University of Chinese Academy of Sciences, Beijing, China. [4] Key Laboratory of Preventive Veterinary Medicine in Hubei Province, The Cooperative Innovation Center for Sustainable Pig Production, Huazhong Agricultural University, Wuhan, China. [5] CAS Key Laboratory of Soft Matter Physics, Institute of Physics, Chinese Academy of Sciences, P.O.Box 603, Beijing, China. [6] Center for Biological Imaging, Institute of Biophysics, Chinese Academy of Sciences, Beijing, China. [7] Departments of Pathology, College of Veterinary Medicine, University of Georgia, Athens, GA, USA. [8] Institute of Molecular and Clinical Medicine, Kunming Medical University, Kunming, China. [9] iHuman Institute, ShanghaiTech University, Shanghai, China. [10] These authors contributed equally: Xiyong Song, Yuejun Shi, Wei Ding. ✉email: liuzhj@shanghaitech.edu.cn; penggq@mail.hzau.edu.cn

Coronaviruses (CoVs) are enveloped, positive-sense, single-stranded RNA viruses with the largest genomes known among RNA viruses (~26–32 kb) belonging to the order *Nidovirales*[1]. *Coronavirinae* is categorized into four genera, *Alpha-*, *Beta-*, *Gamma-* and *Deltacoronavirus* based on sequence alignment of the viral genomes[2], and they cause severe respiratory diseases. To date, seven coronaviruses, namely the alphacoronaviruses HCoV-229E and HCoV-NL63, the betacoronaviruses HCoV-OC43, HCoV-HKU1, SARS-CoV, MERS-CoV and the emerging coronavirus (SARS-CoV-2) circulate globally in the human population accounting for one-third of mild respiratory infection and atypical pneumonia in human[3–5]. SARS-CoV emerged in 2002–2003, and MERS-CoV surfaced in 2012, infecting more than 8000 and 2000 individuals with mortality rates of ~10% and 35%, respectively[4,6,7]. Recently, the ongoing SARS-CoV-2 (severe acute respiratory syndrome coronavirus 2) outbreak has caused a global pandemic, now named COVID-19 (coronavirus disease 2019) by WHO. However, specific human vaccines or antiviral treatments are still under development so far.

The coronavirus spike protein (S protein), an envelope-anchored trimeric type I transmembrane glycoprotein, mediates receptor binding and the fusion of the viral and host cell membranes, which is treated as the main target of neutralizing antibodies and vaccine development[5]. The S protein is composed of two subunits, the N-terminal S1 subunit containing the N-terminal domain (NTD) and receptor-binding domain (RBD), which are responsible for binding cellular receptors (sugars and proteins)[8–17], and the C-terminal S2 subunit possessing a fusion peptide (FP), two heptad repeats (HR) and a transmembrane domain, which drives membrane fusion by undergoing a large conformational rearrangement[18]. Many cryo-EM structures of S glycoprotein trimers in the prefusion conformation have been determined, for instance, mouse hepatitis virus (MHV)[19], human coronavirus HKU1 (HCoV-HKU1)[20], human coronavirus NL63 (HCoV-NL63)[21], SARS-CoV[22,23], MERS-CoV[22], porcine deltacoronavirus (PDCoV)[24,25], infectious bronchitis coronavirus (IBV)[26], human coronavirus OC43 (HCoV-OC43)[9], porcine epidemic diarrhea virus (PEDV)[27], feline infectious peritonitis virus (FIPV)[28], and SARS-CoV-2[29,30]. The S trimer structures of the *betacoronavirus* MHV and HCoV-HKU1 and *gammacoronavirus* IBV display a domain-swapping organization of NTDs and RBDs in the cross-subunit S1 quaternary packing mode[19,20,26], and the exposed (standing) states of the RBDs (of SARS-CoV, MERS-CoV and SARS-CoV-2) that are readily recognized by the receptor have been captured and determined by Cryo-EM methods[22,23,29]. However, the prefusion cryo-EM structures of the *alphacoronavirus* HCoV-NL63 and the *deltacoronavirus* PDCoV S trimers indicate that the RBDs are buried (the lying state) in the intra-subunit S1 quaternary packing mode[21,24–26], suggesting that conformational changes are required to expose the RBDs and render the putative receptor-binding loops available for receptor binding. To date, the crystal structure of the HCoV-229E RBD and aminopeptidase N (APN) complex has been reported[11] and reveals that loops 1–3 (loop 1: residues Phe308-Val325; loop 2: residues Ala352-Arg359 and loop 3: residues Trp404-Lys408) exclusively mediate the interaction with hAPN. However, how the Alphacoronavirus S proteins expose their RBDs for receptor binding remains unclear. Understanding the binding of S trimers in the intra-subunit S1 quaternary packing mode to host receptors and the associated conformational changes is pivotal for the development of antiviral agents against coronaviruses.

In addition, the cleavage mediated by proteases at the S1/S2 and S2′ cleavage sites is critical for membrane fusion[31–33]. In betacoronaviruses, the binding of SARS-CoV S homotrimers to receptors on host cells promotes the release of the S1-ACE2 complex from the S trimer; upon its release, the S2′ trigger loop may be exposed and then cleaved for subsequent fusion activation[10,34]. The *alphacoronavirus* HCoV-229E exploits trypsin, cathepsin L and TMPRSS2 to complete the fusion activation mediated by the S protein[35–37]. Moreover, fusion activation of HCoV-229E may be highly reliant on cleavage of the S2′ trigger loop ($GSR^{685}{\downarrow}V^{686}AG$)[35]. Recently, the cryo-EM structure of the HCoV-229E spike mutant (dual-proline mutation, Thr871Pro/Ile872Pro) was determined and showed that S protein can expose a portion of its helical core in S2 subunit to solvent, which may facilitate the transition from pre- to postfusion[38]. However, the detailed conformational transition process of the *Alphacoronavirus* S2 subunit by exposing its FP to activate its membrane fusion remains to be further elucidated.

Here, we report the cryo-EM structures of the HCoV-229E S trimers with two conformations, named conformation 1 and 2, at resolutions of 3.21 and 3.55 Å, respectively. Our structures illustrate the dynamic conformational changes that occur in the prefusion states of *Alphacoronavirus* S trimers, providing a better understanding of the molecular mechanisms underlying receptor binding and the transition from pre- to postfusion.

## Results

**Overall structure of the HCoV-229E S trimer**. To structurally characterize the conformation of the HCoV-229E S trimer in the prefusion state, its wild-type soluble ectodomain (residues 1–1116) was expressed as a fusion protein with a C-terminal GCN4 trimerization motif and purified by tandem affinity and gel filtration columns. There were three major peaks in the SEC profile, namely peak 1, peak 2, and peak 3 and the S protein samples from peak 2 and peak 3 were used for further cryo-EM analysis (Supplementary Fig. 1). It is difficult to capture the different conformational changes of RBDs in the prefusion state of the *Alphacoronavirus* S protein from the structures reported previously[21,27,28,38]. Hence, we collected a large amount of cryo-electron microscopy data for searching the different conformational states of the HCoV-229E spike. For the peak 2 sample, a total of 403,347 particles were picked from 2,779 micrographs (Supplementary Fig. 2a) and a total of ~659,410 particles were picked from 5081 micrographs for the peak 3 sample (Supplementary Fig. 2b). After further data processing, we finally identified two major conformational states: conformation 1 (derived from peak 2) and conformation 2 (derived from peak 3) with resolutions of 3.21 and 3.55 Å, respectively (Supplementary Table 1, Supplementary Figs. 3 and 4).

The atomic structures of conformations 1 and 2 (named C1 and C2 hereinafter, respectively) include residues Ile48 to Tyr1033, which cover most of the key structural elements (Fig. 1 and Supplementary Fig. 5). The C-terminal HR2 was not built due to the lack of interpretable density in this region (Fig. 1a). The structures of HCoV-229E S proteins form the overall mushroom-shaped trimers, which are similar to the previously reported structures (MHV, IBV, PDCoV, and HCoV-229E)[19,24–26,38] (Fig. 1b, c and Supplementary Fig. 6). The C1 presents a slender conformation with a total of ~145 Å long and ~113 Å wide, while the C2 is a stubby structure with ~132 Å long and ~117 Å wide (Fig. 1b, c), which is ~13 Å shorter than C1 and the shortest among the S protein structures reported previously (Supplementary Fig. 6). Meanwhile, we observed that the interaction interface between protomers decreased from C1 to C2 (with average interaction interfaces of ~4824 and ~4145 Å² for C1 and C2, respectively).

In addition, through analysis of the C1 density map, there are 31 sugar molecules identified on the surface of the S protein, most of which are located in the S1 subunit. However, only 24 sugar molecules can be traced for the C2 density map due to its

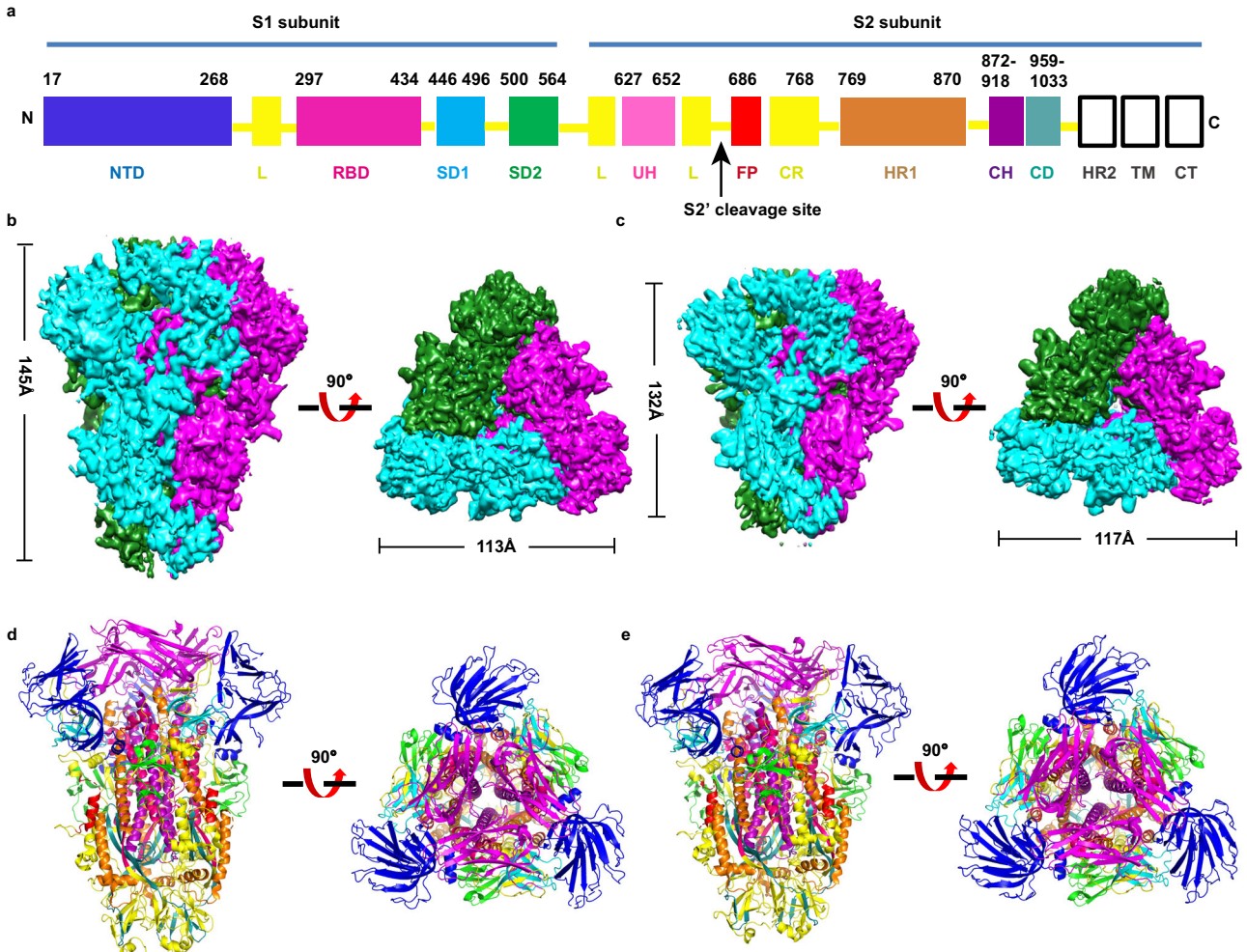

**Fig. 1 Overall structure of the HCoV-229E spike in the prefusion conformation. a** Schematic diagram of the HCoV-229E S glycoprotein organization. S1 receptor-binding subunit, S2 membrane fusion subunit, NTD N-terminal domain, blue, L linker region, yellow, RBD receptor-binding domain, magenta, SD1 subdomain1, cyan, SD2 subdomain2, green, UH upstream helix, pink, FP fusion peptide, red, CR connecting region, yellow, HR1 heptad repeat 1, orange, CH central helix, purple, CD connector domain, aquamarine, HR2 heptad repeat 2, TM transmembrane domain, CT cytoplasmic tail, the range of FP used to make the figures is consistent with that of a previous structural study on the MHV spike[18]. The black line boxes denote the regions that are unresolved in the density map (HR2) or not part of the construct (TM and CT). **b**, **c** The orthogonal views of the S protein in conformations 1 and 2. The protomer A, B, and C are colored as cyan, magenta, and green, respectively. The length of conformations 1 and 2 is measured via the PyMOL. **d**, **e** The atomic structures of conformations 1 and 2 are shown as ribbon diagrams oriented as in (**b**) and (**c**). The domains are indicated and colored as defined in Fig. 1a.

resolution limitation (Supplementary Figs. 5 and 7). Moreover, the overall structures of the S1 regions for C1 and C2 are similar to the counterparts in the other reported coronaviruses with the intra-subunit S1 quaternary packing mode (Supplementary Fig. 8). The β-strand-rich S1 subunit consists of an NTD (residues Ile48-Ser268), RBD (residues His297-Gly434), subunit domain 1 (SD1, residues Cys446-Pro496), and subunit domain 2 (SD2, residues Pro500-Val564) (Fig. 1a, d, e). Similarly, the crown-like trimeric S1 is located at the top of the S2 stalk. There are a large number of hydrophobic interactions between S1 and S2 subunits (Supplementary Fig. 9). The findings indicate that S1 stabilizes the conformation of S2, making it impossible to convert to the postfusion state. Connecting the S1 and S2 subunits are two subdomains, SD1 and SD2, and a long loop. In addition, the α-helix-rich S2 subunit begins at residue Asn568[35], and the atomic model of the S2 subunit includes the functionally important FP (FP, residues Val686-Asp711), heptad repeat 1 (HR1, residues Phe769-Asp870), central helix (CH, residues Ile872-Val918) and connector domain (CD, residues Asp959-Tyr1033) (Fig. 1a). There are differences in the S2 region

between C1 and C2, showing that the length of C2 S2 subunit is ~10 Å shorter than the counterpart of C1 (the RMSD value is 1.10 Å among the 444 Cα atoms) (Supplementary Fig. 10).

**The expanded conformational changes in S1 subunit of the HCoV-229E S trimer may promote its receptor binding.** In our structures, three S1-NTDs are located on the lower and outer sides of S1-RBDs, accounting for a large proportion of the exposed surface area of the S1 subunit. S1-NTD consists of a fourteen-stranded β-sandwich, and the core structure comprises twelve-stranded antiparallel β-sheet layers stacked together through hydrophobic interactions and the overall folds are similar to those of other coronaviruses (Supplementary Fig. 11a). Besides, HCoV-229E S1-RBD adopts a β-sandwich fold containing two β-sheet layers: a three-stranded antiparallel β-sheet and a five-stranded mixed β-sheet (Supplementary Fig. 11b). Similar to the structure of the HCoV-NL63 and PDCoV S trimers[21,24,25], the three NTDs and RBDs of HCoV-229E S1 assume an intra-subunit quaternary packing mode (Fig. 2a). The three S1-RBDs are located at the center and interact with the surrounding

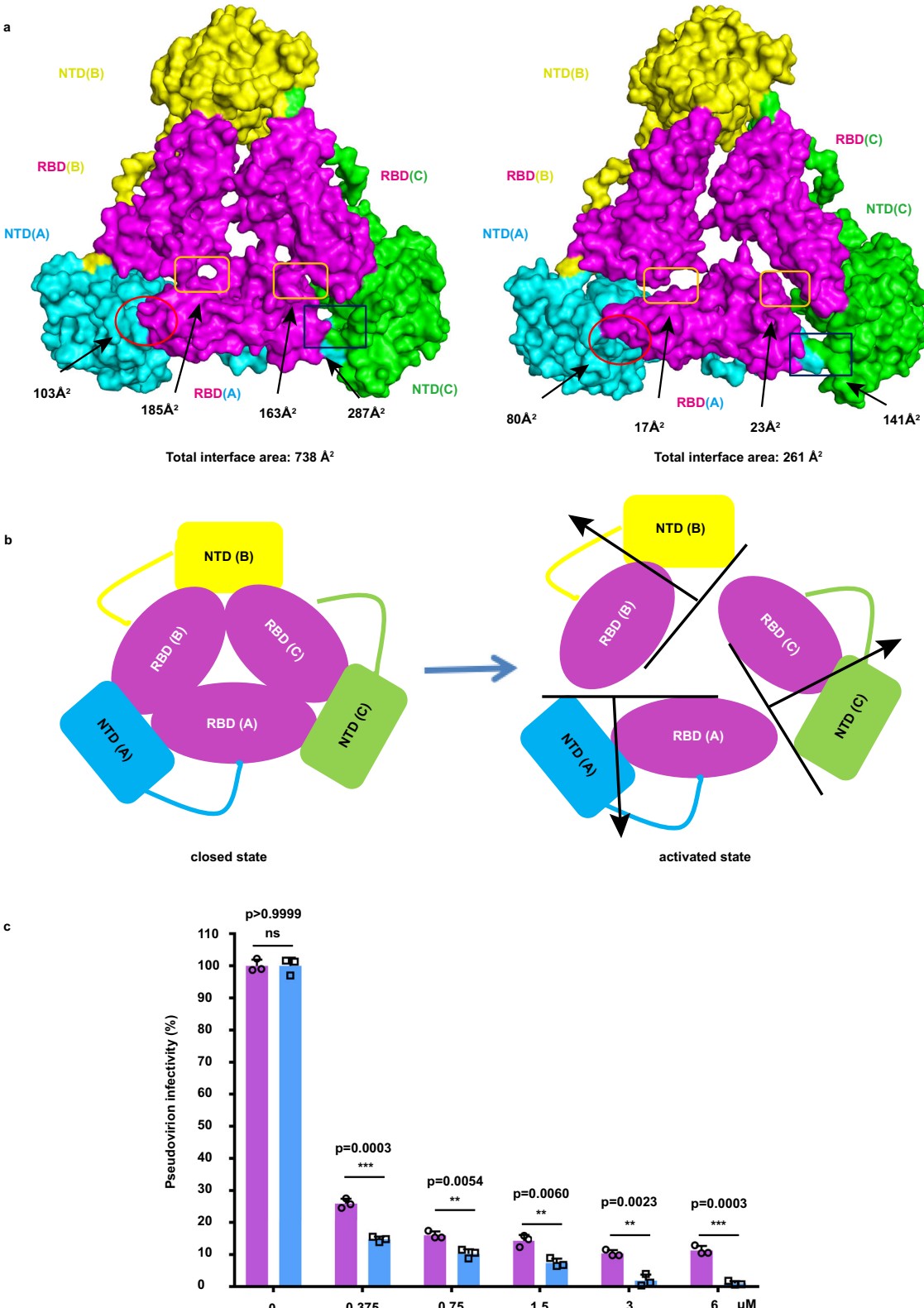

S1-NTDs located on the outer side of the S1-RBDs. The RBD (chain A) interacts with the NTD (chain A) and the adjacent NTD (chain C), RBD (chain C), and RBD (chain B). This interactional mode between the NTDs and RBDs may cause the RBDs in C1 to remain in a lying state (Supplementary Fig. 6). Nevertheless, the S1-RBDs need to transform to a "standing" state to render the putative receptor-binding motif (RBM) loops accessible to the host receptor. Hence, we further analyzed the trimeric structures of C1 and C2 and found that the interaction interface between the S1-RBDs and the surrounding S1-NTDs and RBDs in the adjacent protomers in C2 was smaller than that of C1 (Fig. 2a).

Of note, the S1-RBDs are separated outward from the surrounding NTDs and RBDs, and the interaction interface

**Fig. 2 The expanded conformational changes of the HCoV-229E S protein in S1 subunit. a** The interface areas in the S1 subunit change from C1 to C2. Chains A, B, and C in C1 and C2 are colored cyan, yellow and green, respectively. Besides, the S1-RBDs are colored magenta. The interface areas between the S1-RBD (chainA) and the surrounding S1-NTDs and S1-RBDs from adjacent monomers are shown. The interface areas are analyzed using PDBePISA. **b** A cartoon showing dynamic changes between S1-NTDs and S1-RBDs from C1 to C2 oriented as Fig. 2a. **c** Pseudotyped viral inhibition assay. Huh7 cells were infected with the mixture of HCoV-229E pseudovirus and different S proteins (peaks 2 or 3 samples at the concentration of 0, 0.375, 0.75, 1.5, 3.0, and 6.0 μM). Statistical results were obtained from three independent experiments ($n = 3$ biologically independent samples). Statistical significance was analyzed using an unpaired two-tailed Student's $t$ test. Data are presented as mean values $+/-$ SD (95% confidence interval). The $p$ value was shown. All differences between means with $p < 0.05$ are indicated. *$p < 0.05$, **$p < 0.01$, and ***$p < 0.001$ were considered significant. Source data is provided as source data file. The peaks 2 and 3 samples are colored in magenta and cyan, respectively.

**Table 1 Binding affinities between the spike protein (peaks 2 and 3) samples and hAPN measured using SPR.**

| Spike protein | $K_{on}$ (×$10^4$ $M^{-1}s^{-1}$) | $K_{off}$ (×$10^{-3}$ $s^{-1}$) | $K_D$ (nM) |
|---|---|---|---|
| Peak 2 | 3.18 ± 0.001 | 1.15 ± 0.002 | 36.3 ± 0.071 |
| Peak 3 | 7.15 ± 0.001 | 0.485 ± 0.007 | 6.78 ± 0.105 |

Values after ± correspond to the residual standard deviation. Each experiment was repeated independently twice with similar results.

between the RBD and surrounding NTD and RBDs is decreased from C1 to C2 (with the interface area decreasing from ~635 to ~181 Å$^2$). Meanwhile, the NTD and RBD in the same protomer also separate from each other, and their interface area are decreased (interface area from ~103 to ~80 Å$^2$). The total interface area between the RBD and the other surrounding NTDs and RBDs decreases as well (from ~738 to ~261 Å$^2$). Moreover, the interface area between the RBD and surrounding CH-HR1 junction region (Ile865-Ala874) decreases from ~325 to ~300 Å$^2$ compared C2 with C1 (Supplementary Fig. 9). The total interface area between the RBD and the other surrounding NTDs, RBDs and CH-HR1 helix decreases as well (from ~1063 to ~561 Å$^2$). Thus, we could hypothesize that the conformational transformation from C1 (closed state) to C2 (activated state) leads to fewer interactions in S1 subunit and pose the potential exposure of the S1-RBDs, which favors the receptor binding (Fig. 2b). To address this hypothesis, the interactions between different HCoV-229E S trimers (peaks 2 and 3 samples) and hAPN were tested via the surface plasmon resonance (SPR) experiments and pseudovirus infection assay. In vitro binding measurements showed that the peak 3 sample has a higher hAPN-binding affinity than the peak 2 samples (Fig. 2c). Specifically, the equilibrium dissociation constant ($K_D$) of peak 2 and hAPN is 36.3 ± 0.071 nM, and of peak 3 and hAPN is 6.78 ± 0.105 nM (Table 1 and Supplementary Fig. 12). Consistent with the different binding affinities of peak 2 and peak 3 samples with hAPN receptor, our pseudovirus infection assay results also showed that peak 3 samples could more effectively inhibit viral infection than peak 2 by competitively binding to hAPN in Huh7 cells, which meant that the receptor binding ability of peak 3 (C2) was more effective than that of peak 2 (C1) (Fig. 2c). These findings indicate that the conformational changes of the S1 subunit from a closed state to an activated state cause a decrease in the interface area among the S1-RBDs and S1-NTDs, which may further promote S1-RBDs to bind to hAPN. In summary, our combined results indicate that the S1 subunit of *Alphacoronavirus* S glycoproteins may undergo a dynamic conformational changes before the RBD standing up for their recognition and binding to relative receptors.

**The structural comparison of the S2 subunit between C1 and C2 reveals closed to open dynamic conformational changes.** The coronavirus S2 region is generally considered to be conserved.

However, the overall structural comparison of the S2 subunit of different coronaviruses S proteins revealed that they presented various and dynamic changes (Supplementary Fig. 10). According to the previous research, the open state of the S2 subunit may facilitate the transition of the S trimer from pre- to postfusion[38]. Structural comparison of our two different conformations and the previously reported open HCoV-229E S trimer showed the detailed mechanism of how the S2 subunit gradually opens and exposes its FP before the S2 postfusion conformation is achieved (Fig. 3 and Supplementary Movie 3). Three core helices, namely CH twister outward relative to the C3 symmetry axis from C1 to C2, then to the open HCoV-229E S trimer with twister angles growing from −78° to −86°, resulting in the CH tip in open S trimer moves downward around 5 Å compared with C1 (Fig. 3b). Meanwhile, the triple HR1 helices rotate relative to their tips by ~15° from C1 to C2, then to the open HCoV-229E S trimer, resulting in the Phe in helix (residues Gln791-Phe809) going upward up to ~7 Å for C1 relatively to C2 (Fig. 3c and Supplementary Movie 1). Finally, the FP of the open HCoV-229E S trimer is exposed by 10° rotation compared with C1 (Fig. 3d). The S2 move a little at its top resulting in largely moving at its bottom to further open conformation with FP fully exposed. Therefore, there may be a conformation conversion process from C1 to C2 with some shrinkage (Supplementary Movie 1), and then to the open and stretched HCoV-229E S trimer (Supplementary Movie 2 and Supplementary Movie 3). Besides, in HCoV-229E, the S2′ trigger loop (I$^{676}$PSLPRSGSRVAGR$^{689}$) connects the upstream helix to the FP (residues Val686-Asp711), significantly differing from that of the HCoV-NL63 (L$^{857}$PQRNIRSSRIAGR$^{870}$)[21], PDCoV (L$^{666}$TTRIGGR$^{673}$)[25], SARS-CoV (I$^{787}$LPDPLKPTKR$^{797}$)[22,23], and PEDV (S$^{878}$VYDPASGRVVQ$^{889}$)[27] counterparts (Supplementary Fig. 13a). We observed that the conformation of S2′ trigger loops from C1 and C2 were similar, but the counterpart of the reported open HCoV-229E S protein was obviously shifted out and further exposed (Supplementary Fig. 13b). Hence, we speculate that dynamic conformational changes from C1 to the open HCoV-229E S state may be beneficial to hydrolyzing the S2′ cleavage sites (Arg685) during HCoV-229E entry.

**Discussion**

Cryo-EM analyses showed that the S1 subunits of Alphacoronavirus and Deltacoronavirus S proteins assemble in intra-subunit quaternary packing mode, which makes their S1-RBDs exposure challenging via conformational changes[21,24,25,27,28,38] (Supplementary Fig. 6). Nevertheless, the cross-subunit S1 quaternary packing mode in betacoronaviruses (SARS-CoV, MERS-CoV, and SARS-CoV-2) can allow S1-RBDs to switch to the "standing" conformation for receptor binding without obvious steric clashes[10,22,23,29] (Supplementary Fig. 6). Therefore, how the Alphacoronavirus and Deltacoronavirus S trimers bind receptors via conformational changes remains to be fully elucidated. In this study, our S proteins are the wild-type HCoV-229E S trimers without any mutations at the S1/S2 cleavage sites, which is different from the previous work by introducing the mutations at

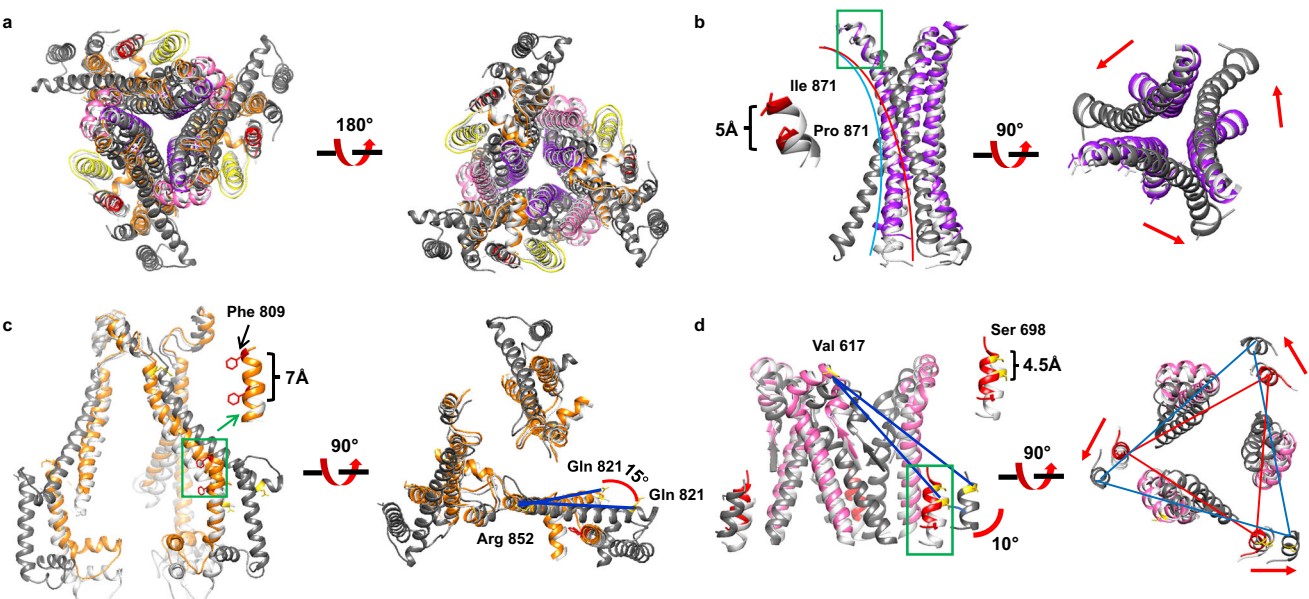

**Fig. 3 Structural comparison of HCoV-229E spike S2 subunit in different conformaitons reveals closed-to-open conformational changes. a** Overall structural comparison of the S2 subunit from C1, C2, and the previously reported open HCoV-229E S protein (PDB ID: 6U7H). **b** Zoomed-in comparison of CH. The CH is twisted outward relatively to the symmetry axis gradually from C1 to C2 to open HCoV-229E S protein with twister angles of −78°, −78.2° and −86°, respectively. **c** Close-up views of HR1. The Phe in helix (residues Gln791-Phe809) of C2 goes upward 7 Å compared to that of C1. The helix (residues Gln821-Arg852) of open HCoV-229E S protein rotates outward by 15° Compared to that of C1 (similar to C2). **d** Structural comparison of FP. The FP of C2 goes upward 4.5 Å relatively with that of C1 and can be further exposed with a clockwise rotation of about 10° from C1 to the open HCoV-229E S protein. C1 and the previously reported open S protein (PDB ID: 6U7H) are colored in gray and dark gray, respectively. The C2 is colored as Fig. 1a. Some loops between the secondary structures are omitted for clarification.

the S1/S2 regions to enhance sample stability and homogeneity[38]. This could be the reason we found the S protein adopts closed C1 and activated C2 simultaneously. A recent research work for the SARS-CoV-2 S protein showed similar results that the dominant S protein presented in the tightly closed state and only a minor is in the open state[39]. We superimposed the RBD-hAPN complex crystal structure[11] onto RBD of the C2 in which all RBDs are in the "lying" positions and found that numerous steric clashes were observed between the hAPN and RBD. But the SPR experiments showed that both C1 and C2 can bind to the hAPN receptor, which is similar to the fact that the closed S1 subunit of HCoV-NL63 and SARS-CoV-2 can bind to their ACE2 receptors[21,30]. In addition, the peak 3 sample can block the infection of the pseudovirus to its host cells more efficiently than that of peak 2 samples, which is consistent with the SPR results that the C2 has a higher affinity to the hAPN receptors than C1. Taken together, the C2 function well both in vitro and in vivo. Besides, the reason why C2 can function well could be explained that the hAPN receptors induce the RBDs of C2 transit to "standing" positions to bind the receptor. This similar phenomenon was also found for SARS-CoV-2 S protein[39]. The presence of ACE2 can greatly shift the population landscape of S trimer from the closed conforamtion (94%) to the RBD standing open state (26.2% of ACE2 free and 73.8% of ACE2 bound)[39]. Our work provides the clues on how the RBD domains of SARS-CoV-2 S proteins stand up capable of the receptor binding which could be used for the corresponding antiviral drugs design to stabilize the S protein in the prefusion state.

We could speculate that the C1 can transit to C2. In the C1 state, S1-RBDs are buried through their interactions with the surrounding S1-NTDs and S1-RBDs in the same or adjacent protomer and are not available for binding to hAPN (Fig. 2a). The interactions between the S1-RBDs and the surrounding S1-NTDs and S1-RBDs in the C2 state are decreased with an

expanded conformation presented in the S1 subunit, which may indicate that the RBD is much easier to transform to the "standing" conformation because a lower energy barrier needs to be overcome (Fig. 2a, b). This could be confirmed by the fact that the activated C2 can bind to hAPN receptor with higher affinity tested by our SPR experiments and pseudovirus infection assay (Fig. 2c and Table 1). Though the mechanisms of how S1-RBD transforms to the standing state remains to be further elucidated since this standing state has not been captured until now. However, we still could propose that the RBDs from inaccessible S protein closed state (C1) might be transformed to the activated state (C2) and then to the "standing" conformation for the receptor binding (Figs. 2a, b and 4). Hence, we propose that the activated state C2 may be the intermediate state of RBD transition to the "standing" state (Fig. 4); however, the "standing" state of the RBD may be dynamic and unstable in *Alphacoronavirus*, which is not easy to capture[21,27,28,38]. Since *Alphacoronavirus* and *Deltacoronavirus* have similar intra-subunit quaternary packing modes[26], we could speculate that the closed to activated state transition observed in HCoV-229E may also occur in the S proteins of other alphacoronaviruses and deltacoronaviruses.

In C1 and C2, three S1 subunits form a cap that sits over the S2 subunits and prevent the S2 subunits from a pre- to postfusion state transition, similar to the previous studies[9,10,18–30,34,38,40] (Supplementary Figs. 6 and 9). The initiation of the fusion reaction requires the disassembly of the S protein by cleavages at the S1/S2 and S2′ sites and exposure of FP for binding to the host cell membrane[41,42]. Our two conformational structures combined with the previously reported structure of the HCoV-229E S protein revealed the gradual opening of the S2 subunit with FP exposed by ~10° rotation (Fig. 3). Previous work, including MHV[43,44] and SARS-CoV[34] showed that receptor binding to the S protein can induce its conformational change, exposing the cleavage site for the proteases. This can explain why the C2

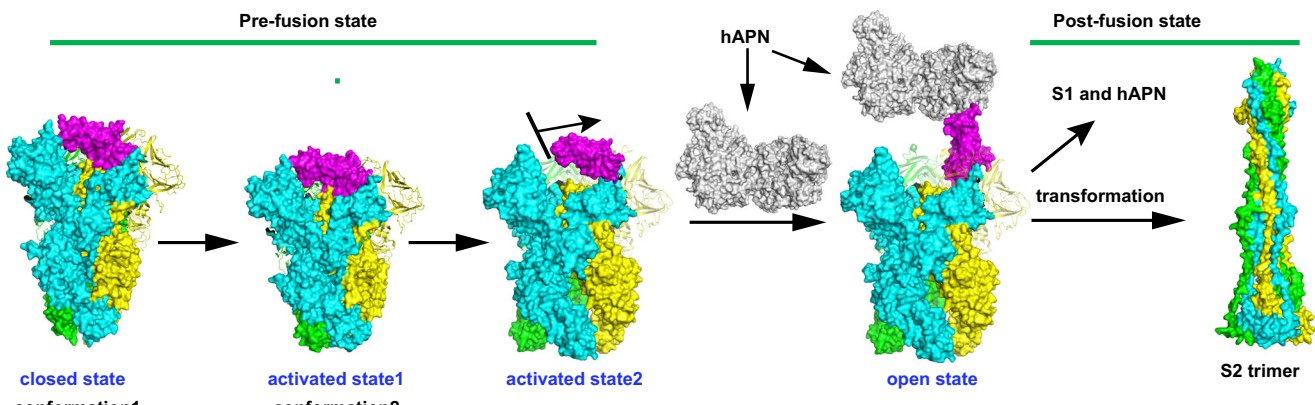

**Fig. 4 Proposed pre- to post-fusion transition model of the HCoV-229E S trimer.** The activated state 2 model and open state model were generated by superimposing the RBD-hAPN complex crystal structure (PDB ID 6ATK) onto the previously reported HCoV-229E structure (PDB ID 6U7H) with some modifications. And the structural model of the S2 trimer in the post-fusion state was predicted using SWISS-MODE based on the homologous structure of the MHV S2 trimer (PDB ID 6B3O). The HCoV-229E S trimer is depicted with each protomer in cyan, yellow and green, respectively. The S1-RBD in protomer and hAPN are colored as magenta and gray, respectively.

presents a small opening conformational change compared with the C1 due to the stabilizing effect of S1 on S2 and lack of hAPN binding. Determining the complex structure of HCoV-229E S trimer and hAPN can further reveal how the receptors binding to the RBDs impacts the conformational change in S2 subunit.

In the intra-subunit S1 quaternary packing mode, *alphacoronavirus* (HCoV-NL63, PEDV, and FIPV)[21,27,28] and *deltacoronavirus* (PDCoV)[25] are capable of shielding vulnerable sites via glycan masking, similar to HIV-1[45]. In our structures, the HCoV-229E spike protein presents compact structure with a large number of glycans (Supplementary Fig. 7). Although the C2 is in an activated state, its S1-RBDs are in a lying state that decreases the exposed surface area compared with that of the standing S1-RBDs state in the betacoronaviruses (SARS-CoV, MERS-CoV, and SARS-CoV-2)[22,23,29] (Supplementary Fig. 6) and potentially limit the accessibility of the RBDs to neutralizing antibodies. A structure-based design strategy has been applied to increase expression and elicit potent antibody responses in the S proteins of betacoronaviruses SARS-CoV and MERS-CoV[40] and in the respiratory syncytial virus fusion (F) glycoprotein[46]. Obtaining the stable prefusion conformation by exposing S1-RBDs via mutagenesis could be a major challenge for the structure-based design of vaccines for alphacoronaviruses in the future.

In summary, we performed detailed 2D and 3D classification for the collected cryo-EM data, but found no "standing" state of RBD and postfusion conformations of S2, which showed that the HCoV-229E S-trimer was a compact structure. Moreover, due to the stabilizing effect of S1 on S2 subunit (Supplementary Fig. 9), the conformational transition from pre- to postfusion is difficult. Receptor binding and proteolytic activation can remove the restriction and promote conformational transformation[34,43,44]. Hence, we propose a feasible membrane fusion process for HCoV-229E as shown below (Fig. 4). First, the inaccessible closed state (C1) can transform to an activated state1 (C2), which reduces the steric conflicts for RBD transformation to the "standing" state. Then, S2 subunit further transforms to an open state. The open state may promote the ability of the S1-RBDs to form a transient "standing" state that exposes RBM loops for receptor binding[11]. Furthermore, S trimers binding to hAPN may facilitate partial dissociation of the S1 subunit from the S2 subunit[34,43,44] with the S2′ cleavage sites exposed and cleaved. Once the S1 subunit is dissociated from the S2 subunit, the heptad repeat 1 alpha-helixes and connecting loops are refolded and reoriented to append to the N terminus of the central helix,

leading to the formation of a long helix that assembles as a homotrimeric helical bundle to insert the FP into the host cell membrane[18], thus pulling the viral and host membranes together. The transition of the HCoV-229E S trimer from pre- to postfusion is thus achieved.

## Methods

**Plasmid construction.** An insect codon-optimized sequence encoding the HCoV-229E S glycoprotein ectodomain (GenBank accession number NP_073551.1, residues 1-1,116) was cloned into the baculovirus transfer vector pFastbac1 (Invitrogen) with a gene fragment encoding a GCN4 trimerization motif (LIKRMKQIEDKIEEIESKQKKIENEIARIKKIK)[18,21,24–26], a thrombin-cleavage site (LVPRGSLE), an eight-residue Strep-tag (WSHPQFEK) and a stop codon. The construct was validated by DNA sequencing. The S sequence of HCoV-229E was synthesized by GenScript Corporation (GenScript, Nanjing, China).

**Protein expression and purification.** The S glycoprotein ectodomain was expressed and purified using a previously described protocol[17]. The construct was transformed into bacterial DH10Bac competent cells (Invitrogen); then, the extracted Bacmid was transfected into Sf9 cells (source: American Type Culture Collection) using Cellfectin II Reagent (Invitrogen). The passage 1 (P1) baculoviruses were harvested and amplified to generate a high-titer virus stock with Sf9 cells, and then they were used to produce the recombinant proteins. The supernatant of the cell culture containing the secreted S glycoprotein was harvested at 60 h after infection and concentrated, and the buffer was changed to binding buffer (10 mM HEPES, pH 7.2, 500 mM NaCl). Finally, S glycoprotein was captured by StrepTactin Sepharose High Performance (GE Healthcare) and eluted with 10 mM D-desthiobiotin in binding buffer. Oligomerization of the HCoV-229E S trimer (1 mg) was analyzed using a Superose 6 Increase 10/300 GL column (GE Healthcare) with a buffer containing 10 mM HEPES, pH 7.2, and 150 mM NaCl[23] at a flow rate of 0.3 ml/min (4 °C). We found three peaks in the gel filtration profile (Supplementary Fig. 1a). Because peak 1 represents a highly aggregated protein, we collected the fractions of peaks 2 and 3 for the cryo-EM analysis. The proteins collected in the different fractions were analyzed by SDS-PAGE.

**Cryo-EM data collection and processing.** Four-microliter samples at a concentration of 0.75 mg/ml for the S trimer proteins (peaks 2 and 3) were applied to a glow-discharged holey carbon grid (Quantifoil, R1.2/1.3, Ted Pella). The grids were blotted using Vitrobot Mark IV (ThermoFisher, USA) with a 4-s blotting time, a force level of 2 at 100% humidity and 4 °C and were then immediately plunged into liquid ethane cooled by liquid nitrogen. Micrographs of the S-protein samples were recorded using a 300 kV Titan Krios G2 electron microscope (ThermoFisher, USA) equipped with a K2 Summit direct electron detector (Gatan, USA) in super-resolution mode with a pixel size of 1.40 Å. Each movie was exposed for 10 s and dose-fractioned into 38 frames with a total dose of ~60 e-/A$^2$ on the samples. The defocus values used during data collection varied from −2.0 to −3.0 μm. All images were collected using the SerialEM automated data collection software package[47].

All images and particles were processed in the platform of cryoSPARC[48,49]. The images were first motion-corrected by "Full-frame motion correction" and their contrast transfer functions were estimated by CTFFIND4[50]. The particles were auto-picked using Template picker and extracted with a box size of 200 pixels. For

peak 2 sample, the auto-picked particles (403,347) were screened and selected by reference-free 2D Classification, while the classes with resolutions below 6 Å would be deleted. The remaining particles (205,567) were then used for ab initio reconstruction (3D classifications) without imposing any symmetry. Among the three classes, one class with S1 and S2 subunits was clearly recognized as C3 symmetry and used for further reference-free 2D Classification. After that, 36,846 particles were selected for final Non-uniform 3D Refinement with C3 symmetry. The density map of conformation 1 was obtained at a resolution of 3.21 Å, based on the 0.143 criterion in the gold standard Fourier Shell correlation coefficient. The data for the peak 3 sample were processed in a similar manner. The auto-picked particles (659,410) were further screened and selected in reference-free 2D Classification step. A total of 244,949 good particles were used for Ab Initio Reconstruction without imposing any symmetry. Two classes with S1 and S2 subunits were clearly recognized as C3 symmetry, but the density map of the first class is very similar to conformation 1 (solved from peak 2), we only used the second class for further 2D Classification and Hetero Refinement. Then, 55,513 particles were left for final Non-uniform 3D refinement with C3 symmetry applied. Based on the 0.143 criterion in the gold-standard Fourier Shell correlated coefficient, the density map of conformation 2 was obtained at a resolution of 3.55 Å. The local resolution of the final density maps of conformation 1 and 2 were analyzed and estimated by the Local Resolution Estimation tool in cryoSPARC.

**Model building and structural analysis**. The homologous cryo-EM structure of HCoV-NL63 S trimer[21] was manually fitted into the corresponding maps of conformations 1 and 2 using CHIMERA[51]. Further improvement of the initial models was processed by iterative positional and B-factor refinement using Phenix real space refinement[52]. The final models were corrected and rebuilt in COOT[53], and evaluated by Phenix Validation Cryo-EM and EMRinger[52]. The buried surface area and the root mean square deviation (RMSD) were analyzed using PDBePISA (http://pdbe.org/pisa/) and CHIMERA, respectively. The amino acid sequences of the coronavirus S glycoprotein were aligned using ClustalW2[54] and visualized with the ESPript 3 server (http://espript.ibcp.fr)[55]. In addition, the NCBI accession numbers of the sequences used were as follows: HCoV-229E (GenBank ID: NP_073551.1), HCoV-NL63 (GenBank ID: Q6Q1S2), PDCoV (GenBank ID: KT336560), PEDV (GenBank ID: KC140102.1) and SARS-CoV (GenBank ID: NP_828851.1).

**Pseudotyped-virus infection assay**. The HCoV-229E pseudovirus entry assay was carried out as previously described[37]. The full-length HCoV-229E S gene was inserted into the pcDNA3.1 (+) plasmid. Retroviruses pseudotyped with the HCoV-229E S expressing a luciferase reporter gene were prepared by co-transfecting HEK293T cells (American Type Culture Collection) with a plasmid carrying the Env-defective, luciferase-expressing HIV-1 genome (pNL4-3.luc.RE) and a plasmid encoding HCoV-229E S. The produced HCoV-229E pseudoviruses were harvested at 72-h post transfection and then used for the entry assay in Huh7 cells (American Type Culture Collection).

Huh7 ($1 \times 10^5$) cells were seeded into 48-well plates and incubated until the cells reached ~80% confluence. For the inhibition assays, the S trimer (peaks 2 and 3 at the concentration of 0, 0.375, 0.75, 1.5, 3.0, and 6.0 μM, diluted in PBS) was incubated for 1 h at 37 °C after addition to the cells. The medium was removed, and the cells were inoculated with an equal amount of HCoV-229E pseudovirus (25 μl; $1 \times 10^5$ TCID$_{50}$/ml) and incubated for 24 h at 37 °C. The cells were then washed with PBS and lysed. Aliquots of cell lysates were transferred to an Optiplate-96 (PerkinElmer), followed by the addition of luciferase substrate. Relative light units (RLUs) were measured using an EnSpire plate reader (PerkinElmer). The inhibitory effect was depicted as a percentage relative to the control cells. All measurements were carried out in quadruplicate.

**Surface plasmon resonance**. Binding kinetics of purified peaks 2 and 3 samples to the hAPN were measured by surface plasmon resonance (OpenSPR, Nicoyalife), as described previously[56–58]. In brief, the hAPN (200 μl, 5 μg) was immobilized on the OpenSPR™ COOH Sensor Chip (Nicoya # SEN-AU-100-12-COOH) at a flow rate of 20 μl/min in HBS-EP+ Buffer (GE Healthcare). Free activated carboxyl groups were deactivated with the addition of 100 μl blocking buffer (Nicoya). Then, the immobilized protein was washed with HBS-EP+ Buffer. After achieving a stable baseline, the running buffer was injected for blank measurement followed by successive injections of buffer matched peak 2 (0, 6.25, 12.5, 25, and 50 nM) and peak 3 (0, 3.13, 6.25, 12.5 and 25 nM) at 20 μl /min, and the binding time was 240 s and the natural dissociation 180 s was carried out. Response unit (RU) values were measured at 298 K. Binding kinetic parameters were obtained by fitting the curve to a one-to-one binding model using the TraceDrawer software package (Ridgeview Instruments, Uppsala, Sweden). All injections were carried out in duplicate and gave essentially identical results. Only one of the duplicates is shown.

**Statistical analysis**. Statistical analysis was carried out using GraphPad Prism 7.0. Statistical analysis was conducted on data from three independent experimental replicates. Statistical significance was determined using an unpaired two-tailed Student's t test. Data are presented as mean values +/− SD (95% confidence interval). All differences between means with $p < 0.05$ are indicated. *$p < 0.05$, **$p < 0.01$, and ***$p < 0.001$ were considered significant.

## Data availability

The data that support this work are available from the corresponding author upon reasonable request. EM maps have been deposited in EMDB with accession codes EMD-9744 (conformation 1) and EMD-9745 (conformation 2). In addition, the modeled atomic coordinates have been deposited in the Protein Data Bank with the accession code 6IXA (conformation 1) and 6IXB (conformation 2). Source Data is provided with this article. Source data are provided with this paper.

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

## Acknowledgements

Our cryo-EM work was performed at the Center for Biological Imaging (CBI), Institute of Biophysics, Chinese Academy of Sciences. We would like to thank Fei Sun, Gang Ji, Zhenxi Guo, Deyin Fan, Boling Zhu, and Shuoguo Li from the CBI for help in EM data collection. This work was supported by National Natural Science Foundation of China Grants 31722056 and 31702249, National Key R&D Plan of China Grant 2018YFD0500100, China Postdoctoral Science Foundation Grant 2019M662674 and the Huazhong Agricultural University Scientific and Technological Self-innovation Foundation (program no. 2662017PY028).

## Author contributions

G.Q.P. and Z.J.L. conceived of the project and supervised all experiments. Y.J.S., L.M.S., Y.B.T., and Q.Q.X. performed plasmid construction, protein expression, and purification. Y.J.S., L.M.S., and J.L.S. performed biochemical experiments. X.Y.S., Y.C., Y.P.Z., C.Y.C., and X.J.H. performed cryo-EM sample preparation and data collection. X.Y.S. W.D. and T.X.N. performed cryo-EM data processing. W.D. built and refined the model. Y.J.S., X.Y.S., W.D., S.B.X., Z.F.F., G.Q.P., and Z.J.L. analyzed the data. X.Y.S., Y.J.S., W.D., G.Q.P., and Z.J.L.wrote the paper.

## Competing interests

The authors declare no competing interests.
