## [Peer Review File · Nature Communications]

REVIEWER COMMENTS

Reviewer #1 (Remarks to the Author):

This submission considers how alphacoronavirus (229E) spike protein trimers might change conformations so that they bind host cell receptors and initiate virus entry. The study apparently came from an unexpected finding that 229E spike ectodomains are present in two (or three) different stable forms when they are produced and secreted from insect cells. The forms can be separated from each other by SEC. Two of three forms were structurally resolved by cryo-EM, and these two resolved forms are presented and compared in the report. One spike trimer form is slender and compact, and the other form is broader, shorter, and more opened up at the spike monomer interfaces. The central claim is that the 229E spikes transit from compact to opened forms, and that the opened forms are readied for binding to host cell receptors and initiation of virus entry.

The central claim is of some value to the field. For example, the findings will be of interest to those developing vaccines, including for SARS-CoV-2. The findings will be of interest to researchers studying coronavirus entry. There are, however, several deficiencies with the submission, as itemized below.

1. There is only weak and indirect support for the claim that the C2 spike form is preferentially able to bind host cell receptors. Fig. 3C data are ok but by seem insufficient on their own. Further validation of the claims in the paper would require a demonstration that C2 binding to soluble APN exceeds that of C1. This should be considered.
2. The C2 conformation should be discussed in relation to the known 229E RBD:hAPN structure (ref. #10). If C2 is unable to bind hAPN, then how does it block pseudoviruses in Fig 3C? The suggestion that C2 further transits to a form that binds hAPN in the pseudovirus tests is not consistent with a stable C2 form that exists throughout purification of spikes on SEC. The suggest that C2 transits to a hAPN-binding form but apparently not back to the compact C1 form also requires further explanation.
3. The data and text related to Fig 2 appear to be forced into this submission without much regard for the overall emphasis. More importantly, the failure to identify sialic acid binding by 229E does not "indicate that S1-RBDs are the sole receptor binding regions" (line 282). Other host cell attachment factors were not considered and could be operating. The part dealing with sialic acids should either be further justified or omitted from the report.
4. The data and text related to supp fig. 9 appear to be incidental to the overall submission. Also, the results are expectedly negative (individual RBM loops do not interfere with virus entry). There should either be some justification for presenting these data or they should be omitted.
5. There is a need for extensive editing, as text is frequently confusing: lines 38-39; "closed, to activated, to "reported""; needs revision. Lines 58-59 "Besides"; very awkward here. Line 81; "recognized by the receptor has been captured"; needs to be expanded in order to be clear here. Line 99; by "one-step process" authors may be suggesting that S1-S2 cleavage is not necessary, but it is very unclear what they do mean. Needs revision. Lines 115-131 and figures; the constant interchange between "peak2 and peak3" with "C1 and C2" is unnecessarily confusing. Lines 289-292; the sentences here are unjustified extensions of the actual findings and they also awkwardly mix results with inferences. Needs revision. Line 381, "low titer viruses"; is this reference to baculoviruses, if yes, make clear and expand description.

Reviewer #2 (Remarks to the Author):

In this paper, the authors present two new structures of an alphacoronavirus, HCoV-229E in two states, a longer 'closed' state C1 and a shorter & wider 'activated' state C2. The two conformations are described and compared to related structures quite well, though in some cases vague terms and descriptions are used which could be improved (more details below). The followings are my general and specific comments of this paper.

The PDB reports of the reporting structures are below average relative to all the structures at equivalent resolutions in PDB. I suspect that the modeling might not have been done optimally. The maps shown in figure 1B and 1C appear differently from those shown in EMDB. The maps in EMDB look lower resolution than the claimed resolution because there was no apparent side chain density protrusion from the main chain. We note that the deposited maps might not be 'sharpened' optimally and hence may seem lower resolution upon first inspection. I took the liberty of applying the phenix.auto_sharpen method which brings out the higher resolution detail in the deposited maps. Further inspection of the sharpened maps and models could potentially uncover more details about the two conformations and similarities or differences. Whatever published needs to be consistent with the deposited structure to EMDB and PDB. Some examples of side chain densities should be shown in the supplement and would affirm the readers the claimed resolution. Do the maps clearly resolve the sugar molecules? A supplement figure will help to document its resolvability to substantiate the claim.

After examining the deposited maps and structures from the EMDB and PDB, their descriptions of structural differences in two data sets seem accurate, though the twist in the S2 domain does not seem as pronounced as it is made to appear in Fig. 4. Instead, the S1 domain appears to be more spread out in C2, with fewer inter-protein interactions, allowing for more mobility in the RBD. The S1 on the other hand is more compressed/shorter in C2, but it appears to be more of a vertical movement rather than a twist. I would suggest the authors consider making a video to demonstrate the structure difference. At the least, the authors should describe how they aligned the structures in order to see this twist. One could align the structures using Chimera -> Tools -> Structure Comparison -> MatchMaker which aligns the sequence first, then use corresponding residues to align the two structures.

It is unclear what are the differences among three different maps corresponding to conformation 1. How these structures impact the subsequent biochemical assays?

Several functional assays are performed, and the conclusions arrived at seem a bit uncertain. First, 'single loops' in the S1 RBD are tested individually to see if they can bind to the receptor and inhibit viral infection. It is noted that they do not, and that their unique conformation as placed in the entire S1 RBD is required for receptor binding. This test is not too well motivated or clearly interpreted. It is hard to draw any conclusions, or see how it is useful, since the loops on their own indeed are very unlikely to have the required conformation for binding the receptor. However, if described and motivated better, e.g. as potential inhibition of receptor binding with small peptides, this could actually be an interesting and relevant study. Furthermore, the map density in conformation 2 is not continuous in the loop region and thus need to be clarified in the text as to how it would impact the functional interpretation.

Conformations C1 and C2 are separated (by peaks) and it is tested to what degree each solution can prevent infection. The solution with C2 is found to be more effective at binding the receptor. Since the particles in peak 3 are heterogeneous, does it have an effect on the biochemical assay? One assumption is that the two states are stable and a transition between them does not occur during the test. It is later proposed that such transitions can occur to prepare the pre-fusion state for receptor binding, but it remains unclear what drives such a transition or if it does occur at all. Perhaps these are two stable conformations that a trimer can be in, and it happens that one can bind the receptor better than the other.

A hemagglutination assay is also presented, which shows that the HCoC-229E does not bind sialic acid, and hence this may not be a necessary step in receptor binding. The significance of this and relation to the structures presented need further clarification.

Overall, the structures presented in this manuscript will add to the growing knowledge about coronaviruses. However, it is less obvious how this knowledge will impact the understanding of the

SARS-COV2 which attracts a lot of attention from the biological community.

The followings are more specific comments:

Lines 79, 83, 148-150

"packing mode".

- Could it be briefly described what is meant by 'packing mode'? It doesn't seem to be a commonly used term. Perhaps terms like quaternary structure may be more clear and are more commonly used.
- In the figures (Fig 1, S6, S5), the structures/folds are compared and they seem very similar. However again, 'packing mode' is vague and could refer to 'packing' interactions. The structures may be similar, but the interactions could potentially be different, hence again, 'packing mode' may not be an appropriate term and could be misleading.

Line 135

CryoEM map yields density and not electron density. This is a common mistake made by investigators who do not understand the biophysics and should remember to correct the use of this term in their paper report.

Line 156

"The findings indicate that S1 stabilizes the conformation of S2, making it impossible to convert to the postfusion state."

- This seems like an unsupported/vague conclusion. There are not too many interactions between S1 and S2 (this is not clearly shown in the paper figures), and cleavage of a loop between them causes them to dissociate.

Line 162

"There are differences in the S2 region between C1 and C2, showing that the length of C2 S2 subunit is $\sim 10 \text{ \AA}$ shorter than the counterpart of C1"

- Does this mean that the overall height difference between C1 and C2 is due to this change in the S2 region only? i.e. S1 is the same height between C1 and C2?

Figure 2.

- In (A), it is indicated that a 'star' indicates sugar binding sites. However there is only one star for BCoV S1-NTD. It is not clear if this refers to a single sugar binding site or multiple (perhaps the 3 close to residue 160 in panel B). Should there be a star close to the other sugar binding site close to residue 140 pointed out in panel (B)?

Line 196: "To verify whether individual loops have the ability to bind hAPN, a pseudovirus infection assay was performed. However, single loops 1-3 could not inhibit viral infection even at a high concentration"

- The description of this experiment is not too clear: 1) By 'single loops', we assume the loops were isolated/purified (how?). How does binding affect infection?

Line 199: "which indicated that the definite spatial conformation formed by loops 1-3 together is the prerequisite for RBD binding receptor."

- It's still not clear how the previous experiment indicates this spatial conformation is required for binding. It is already clear that the S1 RBD binds the receptor, but it is not clear what else is being stated here in terms of something specific about which loops bind and what their conformation has to be.

Line 205: "Similar to the structure of the HCoV-NL63 and PDCoV S trimers^{21, 24, 25}, the 205 three NTDs and RBDs of HCoV-229E S1 assume an intra-subunit packing mode"

- Again, 'packing mode' is not clear here, and 'intra-' refers to within each protein. 'Inter-' would refer

to other proteins. These interactions are not clearly shown here but from looking at the architecture in 3D, it seems more like there are few 'intra-' interactions but more substantial 'inter-' interactions. Perhaps this should be double checked and described a bit more clearly in terms of within each protein and with other proteins.

- What does 'subunit' refer to in the title of this section and the text? The S1 domain?

Line 222: "decrease of the steric conflict and pose the potential exposure of the S1-RBDs, which favors the receptor binding"

- This is stated a bit awkwardly. It is understandable what it may be trying to say but perhaps can be said a bit more clearly. Steric clashes are unfavorable energetically and should not happen in any state. Perhaps it can be stated in terms of fewer interactions, less

Line 224-229. Interesting result, although it seems both peaks (C1 or C2) inhibit infection to some degree. Is there any way the spikes could go between states C1 and C2 at any time? Any ideas why the difference at high concentration is larger? This is somewhat counter-intuitive.

Line 261, Fig. S10. Only half of the trigger loop in C2 is built (the other half is dotted line). This seems rather unusual – if half can be seen, why is the other half not visible. In light of this, the two loops cannot be fully compared. In Figure S10, the color scheme in C,D is not described.

Line 271. "Cryo-EM analyses have shown that due to the intra-subunit S1 packing mode of Alphacoronavirus and Deltacoronavirus, exposure of S1-RBDs via conformational changes is challenging."

- Seems like a vague and unclear sentence.

Line 287 "the RBD is much easier to transform to the "standing" conformation because less steric conflict needed to be overcome"

- Again, 'steric conflict' seems like a vague and inappropriate term.

Line 334 "Our findings indicate that the conformational transition of C1 to C2 provides a way to escape the host's immune response"

- Even though C2 may be more spread out, this does not mean it is more accessible to antibodies; the conformational change is not that large. It is not clear that this claim is fully supported.

Line 349. Two conformational states were found, C1, and C2. But it is unclear how one can conclude that one state transforms to the other state. If this is really true, then the assays in Figure 3 may seem meaningless because they rely on the fact that each conformation stays the same throughout the experiment.

Supplement table.

It is too low an electron microscope magnification (18.000x) to achieve the reported pixel size of 1.4 Å/pixel?

Reviewer #1 (Remarks to the Author):

This submission considers how alphacoronavirus (229E) spike protein trimers might change conformations so that they bind host cell receptors and initiate virus entry. The study apparently came from an unexpected finding that 229E spike ectodomains are present in two (or three) different stable forms when they are produced and secreted from insect cells. The forms can be separated from each other by SEC. Two of three forms were structurally resolved by cryo-EM, and these two resolved forms are presented and compared in the report. One spike trimer form is slender and compact, and the other form is broader, shorter, and more opened up at the spike monomer interfaces. The central claim is that the 229E spikes transit from compact to opened forms, and that the opened forms are readied for binding to host cell receptors and initiation of virus entry.

The central claim is of some value to the field. For example, the findings will be of interest to those developing vaccines, including for SARS-CoV-2. The findings will be of interest to researchers studying coronavirus entry. There are, however, several deficiencies with the submission, as itemized below.

1. There is only weak and indirect support for the claim that the C2 spike form is preferentially able to bind host cell receptors. Fig. 3C data are ok but by seem

insufficient on their own. Further validation of the claims in the paper would require a demonstration that C2 binding to soluble APN exceeds that of C1. This should be considered.

Agreed, we added the surface plasmon resonance (SPR) experiments *in vitro* to test and compare the binding affinities of peak2 and peak3 with the hAPN receptor as your suggestion. Our SPR experiments showed that the peak3 has a higher hAPN-binding affinity than the peak2 samples. Specifically, the equilibrium dissociation constant (K_D) of peak2 and hAPN is 36.3 ± 0.071 nM, and of peak3 and hAPN is 6.78 ± 0.105 nM (Fig.2c and Supplementary Fig. 12). Consistent with the different binding affinities of peak2 and peak3 samples with hAPN receptor, our pseudovirus assay results also showed that peak 3 samples could more effectively inhibit viral infection than peak2 samples by competitively binding to hAPN in Huh7 cells. Combined with these two experiments *in vitro* and *in vivo* prove that the C2 can more effectively bind to hAPN receptor than the C1 does.

The SPR experiments results have been added in the main text.

2. The C2 conformation should be discussed in relation to the known 229E RBD:hAPN structure (ref. #10). If C2 is unable to bind hAPN, then how does it block pseudoviruses in Fig 3C? The suggestion that C2 further transits to a form that binds hAPN in the pseudovirus tests is not consistent with a stable C2 form that exists throughout purification of spikes on SEC. The suggest that C2 transits to a hAPN-binding form but apparently not back to the compact C1 form also requires further explanation.

Agreed, the RBD adopting the “standing” state is the prerequisite for the hAPN binding and the subsequent membrane fusion. We superimposed the previously determined RBD-hAPN complex crystal structure onto RBD of the C2 in which all RBDs are in the “lying” positions and found that numerous steric clashes were observed between the hAPN and RBDs. But our SPR experiments showed that both C1 and C2 can bind to the hAPN receptor with high affinity, which is similar to the facts that the closed S1 subunit of HCoV-NL63 can bind to its ACE2 receptor tested by ACE2-binding ELISA (Walls A, et al. Nat Struct Mol Biol, 2016) and the closed SARS-CoV-2 S protein also can bind to its ACE2 receptor (Cai YF, et al. Science, 2020). Moreover, the peak3 sample can block the infection of the pseudovirus to Huh7 cells more efficiently than that of peak2 sample, which is consistent with the SPR results that the C2 has higher affinity to the hAPN receptor than C1. Taken together, the C2 is function well *in vitro* and *in vivo*.

The reason why C2 can function well could be explained that the hAPN receptor induce the RBDs of C2 transit to “standing” positions to bind the receptor. In this study, the C2 of HCoV-229E S protein only exists as a transient conformation with minor population in our dataset (Supplementary Fig. 2). This similar phenomenon was also found for SARS-CoV-2 S protein (Xu C, et al. bioRxiv, 2020). Their dataset showed that the majority of the particles (~94%) were in the closed conformation while only 6% of particles adopt the open state (Xu C, et al. bioRxiv, 2020). While the presence of ACE2 could greatly shift the population landscape of S trimer, i.e. 26.2% of particles present unliganded open state and 73.8% of particles are ACE2 bound

open state in the ACE2 present sample (Xu C, et al. bioRxiv, 2020). In addition, we also attempted to analyze the cryo-EM structure of the HCoV-229E S trimer in complex with receptor hAPN complex. Unfortunately, we failed to get the corresponding S-hAPN complex since this complex samples were easy to aggregate.

Besides, our S proteins are the wide-type HCoV-229E S trimer without any mutations at the S1/S2 cleavage sites, which is different from the previous work by introducing the mutations at the S1/S2 regions to enhance sample stability and homogeneity (Li ZJ, et al. eLife, 2019). This could be the reason that we found the S protein adopts C1 and C2 simultaneously. Meanwhile, we could speculate that the C1 can transit to C2. Since in the presence of its hAPN receptor, the RBD of C2 could stand up more efficiently, favorable for the hAPN binding, which could be the reason the activated C2 can bind to hAPN receptor with higher affinity tested by our SPR experiments and pseudovirus infection assay (Fig. 2c, d and 4).

Thanks for your suggestion and we added the explanation regarding your suggestion in the first paragraph of the discussion section.

3. The data and text related to Fig 2 appear to be forced into this submission without much regard for the overall emphasis. More importantly, the failure to identify sialic acid binding by 229E does not “indicate that S1-RBDs are the sole receptor binding regions” (line 282). Other host cell attachment factors were not considered and could be operating. The part dealing with sialic acids should either be further justified or omitted from the report.

Agreed, the data and text related to Fig 2 has been deleted from the manuscript.

4. The data and text related to supp fig. 9 appear to be incidental to the overall submission. Also, the results are expectedly negative (individual RBM loops do not interfere with virus entry). There should either be some justification for presenting these data or they should be omitted.

Agreed, the data and text related to supp fig. 9 has been deleted from the manuscript.

5. There is a need for extensive editing, as text is frequently confusing:
lines 38-39; “closed, to activated, to “reported””; needs revision.

Thanks for your suggestion. This sentence has been changed as “the S trimers trended to open the S2 subunit from the closed conformation to open conformation”

6. Lines 58-59 “Besides”; very awkward here.

Thanks for your suggestion. These two sentences “the emerging SARS-CoV 2 has caused 82,724 confirmed cases of human infections and 3327 deaths in China as reported by China CDC (<http://2019ncov.chinacdc.cn/2019-nCoV/>). Besides, SARS-CoV 2 has recently caused 823,719 confirmed cases of human infections and 40,604 deaths in other 204 countries worldwide (<http://2019ncov.chinacdc.cn/2019-nCoV/global.html>).” have been changed as “the ongoing SARS-CoV-2 (severe acute respiratory syndrome coronavirus 2) outbreak has caused a globally pandemic, now named COVID-19 (coronavirus disease 2019) by WHO.”

7. Line 81; “recognized by the receptor has been captured”; needs to be expanded in order to be clear here.

Thanks for your suggestion. We added “and determined by Cryo-EM methods” after the “recognized by the receptor has been captured” phrase in the manuscript.

8. Line 99; by “one-step process” authors may be suggesting that S1-S2 cleavage is not necessary, but it is very unclear what they do mean. Needs revision.

This item “one-step process” has been deleted from the manuscript.

9. Lines 115-131 and figures; the constant interchange between “peak2 and peak3” with “C1 and C2” is unnecessarily confusing.

Thanks for your suggestion. We described that the conformation 1 (C1) and conformation 2 (C2) are derived from peak2 samples and peak3 samples in lines 132-134.

10. Lines 289-292; the sentences here are unjustified extensions of the actual findings and they also awkwardly mix results with inferences. Needs revision.

Agreed, this sentence was deleted from the manuscript.

11. Line 381, “low titer viruses”; is this reference to baculoviruses, if yes, make clear and expand description.

Thanks for your suggestion. This sentence has been changed as “The passage 1 (P1) baculoviruses were harvested and amplified to generate a high-titer virus stock with Sf9 cells”.

Reviewer #2 (Remarks to the Author):

In this paper, the authors present two new structures of an alphacoronavirus, HCoV-229E in two states, a longer 'closed' state C1 and a shorter & wider 'activated' state C2. The two conformations are described and compared to related structures quite well, though in some cases vague terms and descriptions are used which could be improved (more details below). The followings are my general and specific comments of this paper.

The PDB reports of the reporting structures are below average relative to all the structures at equivalent resolutions in PDB. I suspect that the modeling might not have been done optimally. The maps shown in figure 1B and 1C appear differently from those shown in EMDB. The maps in EMDB look lower resolution than the claimed resolution because there was no apparent side chain density protrusion from the main chain. We note that the deposited maps might not be 'sharpened' optimally and hence may seem lower resolution upon first inspection. I took the liberty of applying the phenix.auto_sharpen method which brings out the higher resolution detail in the deposited maps. Further inspection of the sharpened maps and models could potentially uncover more details about the two conformations and similarities or differences. Whatever published needs to be consistent with the deposited structure to EMDB and PDB. Some examples of side chain densities should be shown in the supplement and would affirm the readers the claimed resolution. Do the maps clearly resolve the sugar molecules? A supplement figure will help to document its resolvability to substantiate the claim.

Thanks for your constructive and excellent suggestions. The two conformational maps have been sharpened using the phenix.auto_sharpen method, which brings out more high resolution details as expected. More side chains are presented than before. The previously deposited maps in EMDB will be superseded by the newly sharpened maps with new EMDB entries of EMD-30496 (C1) and EMD-30497 (C2). The corresponding PDBs were finely modified with the sugar molecules added based on the sharpened maps. Thirty one sugar molecules can be traced in the sharpened C1 map, while twenty four sugar molecules in C2. The representative density maps of S proteins with residues or sugar molecules fitted are shown in supplementary Fig. 5. Besides, the PDBs of C1 and C2 also will be superseded by the new entries of 7CYC (C1) and 7CYD (C2). Searching for the old IDs will be automatically re-direct to the new deposition. Hence, we use the old IDs in our manuscript. Since the newly revised density maps and corresponding PDBs will be released upon publication, please see the revised density maps and PDBs uploaded in the revised systems directly.

After examining the deposited maps and structures from the EMDB and PDB, their descriptions of structural differences in two data sets seem accurate, though the twist in the S2 domain does not seem as pronounced as it is made to appear in Fig. 4. Instead, the S1 domain appears to be more spread out in C2, with fewer inter-protein interactions, allowing for more mobility in the RBD. The S1 on the other hand is more compressed/shorter in C2, but it appears to be more of a vertical movement rather than a twist. I would suggest the authors consider making a video to demonstrate the structure difference. At the least, the authors should describe how

they aligned the structures in order to see this twist. One could align the structures using Chimera -> Tools -> Structure Comparison -> MatchMaker which aligns the sequence first, then use corresponding residues to align the two structures.

Thanks for your constructive suggestions. We really compared the two structures in chimera similar to your suggestions. The two PDBs of C1 and C2 were split first and their sequences are aligned using MatchMaker, then the structures are finally aligned and compared based on the S1 subunits.

The S2 subunits in C2 presents much shorter and opened with a bit twist out compared with C1, which is very similar to the situations of HCoV-229E S opened structure (PDB 6U7H) compared with HCoV-NL63 S trimer (Li ZJ, et al. eLife, 2019). Three videos have been made as your suggestion. Video 1 depicts the structure morphing from C1 (gray) to C2 (hot pink). Video 2 shows the structural morphing from C2 (hot pink) to the opened structure (dim gray, PDB 6U7H). Video3 combines the entire structure rearrangements from C1 to C2 to the opened HCoV-229E S structure (6U7H).

It is unclear what are the differences among three different maps corresponding to conformation 1. How these structures impact the subsequent biochemical assays?

All the C1 are very similar without discernable differences. The only difference is that they have different resolutions, so we selected the C1 derived from peak2 sample as the target to compare with the C2 due to its highest resolution. Due to C1 are the same, the additional C2 in peak3 sample is the new variable impacting the biochemical assay results.

Several functional assays are performed, and the conclusions arrived at seem a bit e. First, 'single loops' in the S1 RBD are tested individually to see if they can bind to the receptor and inhibit viral infection. It is noted that they do not, and that their unique conformation as placed in the entire S1 RBD is required for receptor binding. This test is not too well motivated or clearly interpreted. It is hard to draw any conclusions, or see how it is useful, since the loops on their own indeed are very unlikely to have the required conformation for binding the receptor. However, if described and motivated better, e.g. as potential inhibition of receptor binding with small peptides, this could actually be an interesting and relevant study. Furthermore, the map density in conformation 2 is not continuous in the loop region and thus need to be clarified in the text as to how it would impact the functional interpretation.

Agreed, this part text and data related supplementary Fig.S9 have been deleted as suggested by the reviewer 1.

Conformations C1 and C2 are separated (by peaks) and it is tested to what degree each solution can prevent infection. The solution with C2 is found to be more effective at binding the receptor. Since the particles in peak 3 are heterogeneous, does it have an effect on the biochemical assay? One assumption is that the two states are stable and a transition between them does not occur during the test. It is later proposed that such transitions can occur to prepare the pre-fusion state for receptor binding, but it remains unclear what drives such a transition or if it does occur at all. Perhaps these are two stable conformations that a trimer can be in, and it happens that one can bind the receptor better than the other.

Thanks for your constructive suggestions. Agreed, the conformations C1 and C2 probably are two stable conformations and the factors which drive the transition from C1 to C2 needs to be illustrated in the further researches. Both the RBDs of C1 and C2 need to stand up for binding the hAPN receptor, and the hAPN receptor also could induce this transition in this process, which is similar to the situations of SARS-CoV-2 (Xu C, et al. bioRxiv, 2020).

A hemagglutination assay is also presented, which shows that the HCoC-229E does not bind sialic acid, and hence this may not be a necessary step in receptor binding. The significance of this and relation to the structures presented need further clarification.

Agreed, this part related to the NTD binding sugar molecules has been deleted from the main text as suggested by the reviewer 1.

Overall, the structures presented in this manuscript will add to the growing knowledge about coronaviruses. However, it is less obvious how this knowledge will impact the understanding of the SARS-COV2 which attracts a lot of attention from the biological community.

Agreed, according to the reviewer's suggestion, we have added this part in the second paragraph of the discussion section.

The followings are more specific comments:

Lines 79, 83, 148-150

“packing mode”.

- Could it be briefly described what is meant by ‘packing mode’? It doesn't seem to

be a commonly used term. Perhaps terms like quaternary structure may be more clear and are more commonly used.

Thanks for your suggestion.

According to the reviewer's suggestion, the "packing mode" has been replaced with "quaternary packing mode" based on previous literature report (Shang J, et al. PLoS pathogens, 2018). The "quaternary packing mode" means the way that the S1-NTDs and S1-RBDs in the HCoV-229E S trimer pack together.

- In the figures (Fig 1, S6, S5), the structures/folds are compared and they seem very similar. However again, 'packing mode' is vague and could refer to 'packing' interactions. The structures may be similar, but the interactions could potentially be different, hence again, 'packing mode' may not be an appropriate term and could be misleading.

Thanks for your suggestion.

According to the reviewer's suggestion, the "packing mode" has been replaced with "quaternary packing mode" based on previous literature report (Shang J, et al. PLoS pathogens, 2018). The "quaternary packing mode" means the way that the S1-NTDs and S1-RBDs in the HCoV-229E S trimer pack together.

Line 135

CryoEM map yields density and not electron density. This is a common mistake made by investigators who do not understand the biophysics and should remember to correct the use of this term in their paper report.

Agreed, we displaced the "electron density" with "density" in the main text as

your suggestion.

Line 156

“The findings indicate that S1 stabilizes the conformation of S2, making it impossible to convert to the postfusion state.”

- This seems like an unsupported/vague conclusion. There are not too many interactions between S1 and S2 (this is not clearly shown in the paper figures), and cleavage of a loop between them causes them to dissociate.

Thanks for your suggestion. We added a new figure (Supplementary Fig. 9) in showing the interactions between the S1 and S2 subunits, which were very similar to the situations of the opened HCoV-229E S trimer (Li ZJ, et al. eLife, 2019).

The interactions between the S1 subunits and the S2 subunits of the trimer are caused by several large hydrophobic forces capable of mediating major stabilizing function (Supplementary Fig. 9). Specifically, the SD1 and SD2 domains of each S1 subunit clamp over a hydrophobic knob (residues Ala709-Pro737) that protrudes from the helical core region of each S2 subunit, resulting in around 900 Å² of apolar interaction area (with a total of ~2700 Å²). The RBD domain in S1 subunit makes an additional contact with the HR1-CH junction regions of the helix bundles (residues Ile865-Ala874) in S2 subunit. Taken together, these observations suggest that the S1 cap plays a major role in preventing the prefusion conformational changes required for conversion to the post-fusion form.

Line 162

“There are differences in the S2 region between C1 and C2, showing that the length of

C2 S2 subunit is ~ 10 Å shorter than the counterpart of C1”

- Does this mean that the overall height difference between C1 and C2 is due to this change in the S2 region only? i.e. S1 is the same height between C1 and C2?

Thanks for your suggestion. Both the S1 and S2 subunits account for the shorter C2 than C1. S1 is shortened slightly around 3 Å, while the S2 is condensed by ~ 10 Å (Fig. 1 and Supplementary Fig. 10).

Figure 2.

· In (A), it is indicated that a ‘star’ indicates sugar binding sites. However there is only one star for BCoV S1-NTD. It is not clear if this refers to a single sugar binding site or multiple (perhaps the 3 close to residue 160 in panel B). Should there be a star close to the other sugar binding site close to residue 140 pointed out in panel (B)?

Thanks for your suggestion. According to the reviewer 1’s suggestion, this part has been deleted from the manuscript.

Line 196: “To verify whether individual loops have the ability to bind hAPN, a pseudovirus infection assay was performed. However, single loops 1-3 could not inhibit viral infection even at a high concentration”

· The description of this experiment is not too clear: 1) By ‘single loops’, we assume the loops were isolated/purified (how?). How does binding affect infection?

Thanks for your suggestion. According to the reviewer 1’s suggestion, this part has been deleted from the manuscript.

Line 199: “which indicated that the definite spatial conformation formed by loops 1-3 together is the prerequisite for RBD binding receptor.”

· It's still not clear how the previous experiment indicates this spatial conformation is required for binding. It is already clear that the S1 RBD binds the receptor, but it is not clear what else is being stated here in terms of something specific about which loops bind and what their conformation has to be.

Thanks for your suggestion. According to the reviewer 1's suggestion, this part has been deleted from the manuscript.

Line 205: "Similar to the structure of the HCoV-NL63 and PDCoV S trimers^{21, 24, 25}, the

205 three NTDs and RBDs of HCoV-229E S1 assume an intra-subunit packing mode"

· Again, 'packing mode' is not clear here, and 'intra-' refers to within each protein. 'Inter-' would refer to other proteins. These interactions are not clearly shown here but from looking at the architecture in 3D, it seems more like there are few 'intra-' interactions but more substantial 'inter-' interactions. Perhaps this sound be double checked and described a bit more clearly in terms of within each protein and with other proteins.

· What does 'subunit' refer to in the title of this section and the text? The S1 domain?

Thanks for your suggestion.

According to the reviewer's suggestion, the "packing mode" has been replaced with "quaternary packing mode" based on previous literature report (Shang J, et al. PLoS pathogens, 2018). The "quaternary packing mode" means the way that the S1-NTDs and S1-RBDs in the HCoV-229E S trimer pack together.

Besides, the S1 subunit covers the NTD, RBD, SD1 and SD2 subdomain, which

was illustrated in Fig. 1A. Also, the previous research work usually divided the S protein into S1 and S2 subunits. And the former is responsible for the receptor binding and the latter for the membrane fusion.

Line 222: “decrease of the steric conflict and pose the potential exposure of the S1-RBDs, which favors the receptor binding”

· This is stated a bit awkwardly. It is understandable what it may be trying to say but perhaps can be said a bit more clearly. Steric clashes are unfavorable energetically and should not happen in any state. Perhaps it can be stated in terms of fewer interactions, less

Agreed, the “decrease of the steric conflict” has been changed as “leads to fewer interactions in S1 subunit” based on your suggestion.

Line 224-229. Interesting result, although it seems both peaks (C1 or C2) inhibit infection to some degree. Is there any way the spikes could go between states C1 and C2 at any time? Any ideas why the difference at high concentration is larger? This is somewhat counter-intuitive.

The C1 could transit to C2 to decrease the interactions between the protomers, especially in the S1 subunit, which is favorable for the exposure of RBD to the hAPN receptor. This speculated sequential transition also was confirmed by the fact that C2 has higher affinity to the hAPN receptors tested by the SPR and pseudovirus infection assay.

For your second question, the possible explanation is as follows. Since the peak3 samples can more effectively inhibit the viral infection with smaller K_D than peak2.

When both the concentrations of peak2 and peak3 samples are far lower than their corresponding saturated inhibitory concentration, the respective inhibitory efficiencies remain small different. But the inhibitory efficiency of these two samples would be magnified with the increasing concentrations used in the experiments since the peak3 samples bind to the hAPN receptors more strongly with only one-sixth K_D value with peak2 samples.

Line 261, Fig. S10. Only half of the trigger loop in C2 is built (the other half is dotted line). This seems rather unusual – if half can be seen, why is the other half not visible. In light of this, the two loops cannot be fully compared. In Figure S10, the color scheme in C,D is not described.

Yes, only half of the trigger loop in C2 was visible. We speculated that this loop is more flexible than that of C1, which maybe the reason that we didn't resolve it in our density map. We aligned the structures of C1 and C2 based on the S1 subunit and then presented the structural alignment of the two trigger loop regions. We deleted the density maps of the trigger loops

The color scheme was added in Supplementary Fig. 13c as your suggestion.

Line 271. “Cryo-EM analyses have shown that due to the intra-subunit S1 packing mode of Alphacoronavirus and Deltacoronavirus, exposure of S1-RBDs via conformational changes is challenging.”

· Seems like a vague and unclear sentence.

Agreed, this sentence has been change as “Cryo-EM analyses showed that the S1 subunits of Alphacoronavirus and Deltacoronavirus S proteins assemble in

intra-subunit quaternary packing mode, which makes their S1-RBDs exposure challenging via conformational changes”.

Line 287 “the RBD is much easier to transform to the “standing” conformation because less steric conflict needed to be overcome”

· Again, ‘steric conflict’ seems like a vague and inappropriate term.

Agreed, the item of “less steric conflict” has been changed as “lower energy barrier”.

Line 334 “Our findings indicate that the conformational transition of C1 to C2 provides a way to escape the host's immune response”

· Even though C2 may be more spread out, this does not mean it is more accessible to antibodies; the conformational change is not that large. It is not clear that this claim is fully supported.

Agreed, this sentence has been deleted from the discussion section.

Line 349. Two conformational states were found, C1, and C2. But it is unclear how one can conclude that one state transforms to the other state. If this is really true, then the assays in Figure 3 may seem meaningless because they rely on the fact that each conformation stays the same throughout the experiment.

Agreed, the C1 and C2 probably are two stable conformations and the factors which drive the transition from C1 to C2 needs to be illustrated in the further researches. For the SPR and pseudovirus infection assay, both C1 and C2 function well *in vitro* and *in vivo*, which indicates that the RBDs were induced to the standing state to complete the binding to the hAPN receptor. This phenomenon is similar to the

situations of SARS-CoV-2 (Xu C, et al. bioRxiv, 2020).

This explanation has been added in the discussion section.

Supplement table.

It is too low an electron microscope magnification (18.000x) to achieve the reported pixel size of 1.4 Å/pixel?

The data was collected using Titan Krios microscopy equipped with standard K2 alone without GIF. So, the magnification of 18,000 really corresponds to the pixel size of 1.4 Å/pixel mentioned in the table 1.

REVIEWERS' COMMENTS

Reviewer #1 (Remarks to the Author):

This resubmission is significantly improved. Questionable and negative data that had not advanced the research were removed, and now a more coherent summary of findings is in place. The work shows two conformations of the HCoV-229E spikes, with details indicating that an open conformation has fewer inter-subunit contacts between NTD and RBD, and possibly exposure of receptor binding sites. A reasonable hypothesis is put forward that suggests an open spike conformation as an intermediate in spike protein-directed virus entry. Textual clarity is also improved in the resubmission, although there are still a few minor comments, itemized below.

1. Title, "prefusional" may not be the right word. "pre-fusion" is used but not "prefusional".
2. Rename "peak 2" as C1 and "peak 3" as C2 throughout the figures and text. This will increase clarity.
3. Lines 212-216; is it clear that 229E changes differ from that of SARS, MERS, SARS-2? Could it be that the experiments here revealed a form of spike that also is part of the SARS, MERS and SARS-2 dynamics, but not yet picked up in others' experiments? Authors' statements in the discussion section seem to argue for this possibility.
4. Line 259, wild, not wide
5. The long discussion section could be made shorter with additional text editing.

Reviewer #2 (Remarks to the Author):

The revised manuscript has taken into account of the reviewers comments. I recommend to seek an editorial checkup on a few typos and grammatical expression throughout the text.

Reviewer #1 (Remarks to the Author):

This resubmission is significantly improved. Questionable and negative data that had not advanced the research were removed, and now a more coherent summary of findings is in place. The work shows two conformations of the HCoV-229E spikes, with details indicating that an open conformation has fewer inter-subunit contacts between NTD and RBD, and possibly exposure of receptor binding sites. A reasonable hypothesis is put forward that suggests an open spike conformation as an intermediate in spike protein-directed virus entry. Textual clarity is also improved in the resubmission, although there are still a few minor comments, itemized below.

1. Title, “prefusional” may not be the right word. “pre-fusion” is used but not “prefusional”.

Thanks for your suggestion. The “prefusional” has been replaced by “prefusion” in the title.

2. Rename “peak 2” as C1 and “peak 3” as C2 throughout the figures and text. This will increase clarity.

Thanks for your suggestion. In the cryo-EM data processing of peak3 sample, we found two conformations (C1 and C2) (Supplementary Figure 2b). Hence, “peak3” are not equivalent to “C2”. To accurately describe our results, the “peak2”, “peak3”, “C1” and “C2” are used in the revised manuscript.

3. Lines 212-216; is it clear that 229E changes differ from that of SARS, MERS, SARS-2? Could it be that the experiments here revealed a form of spike that also is part of the SARS, MERS and SARS-2 dynamics, but not yet picked up in others’

experiments? Authors' statements in the discussion section seem to argue for this possibility.

Agreed, structural and functional studies of coronaviruses S glycoproteins indicate that the S1 subunit would undergo a dynamic conformational changes before the RBD standing up for their recognition and binding to relative receptors. This possibility that the similar conformational changes of HCoV-229E S-trimer may exist in SARS-CoV, MERS-CoV, SARS-CoV-2, so we deleted the phrase “that differ from that of SARS-CoV, MERS-CoV and SARS-CoV-2 reported previously^{22, 23, 29}” for accuracy. In addition, we mentioned a recent work for the SARS-CoV-2 S protein elucidated similar mechanisms in S1 subunit conformation dynamic. We added a sentence “A recent research work for the SARS-CoV-2 S protein showed similar results that the dominant S protein presented in the tightly closed state and only a minor is in the open state.” In the first paragraph of discussion section.

4. Line 259, wild, not wide

The “wide” has been changed as “wild” as your suggestion.

5. The long discussion section could be made shorter with additional text editing.

Thanks for your suggestion, according to the reviewer's suggestion, the discussion section has been shortened in the revised manuscript.

Reviewer #2 (Remarks to the Author):

The revised manuscript has taken into account of the reviewers comments. I recommend to seek an editorial checkup on a few typos and grammatical expression

throughout the text.